# A Combined Surgical Approach for the Treatment of a Sino-Naso-Orbital Glomangiopericytoma

**DOI:** 10.3390/reports6010007

**Published:** 2023-02-15

**Authors:** Ania Labouchère, Cécile De Sandre, Martin Broome, Antoine Reinhard, Aurélien Louvrier

**Affiliations:** 1Department of Maxillofacial Surgery, Lausanne University Hospital, Lausanne University, 1011 Lausanne, Switzerland; 2Department of Otorhinolaryngology-Head and Neck Surgery, Lausanne University Hospital, 1011 Lausanne, Switzerland

**Keywords:** glomangiopericytoma, endoscopy, ENT surgery, maxillo-facial surgery

## Abstract

Glomangiopericytomas are a rare type of sinonasal tumors with high relapse rates. Endoscopic sinus surgery is the classical technique to remove such tumors. Here we report a case of sinonasal glomangiopericytoma with orbital extension, infrequently described in the literature. A 50 year-old woman was referred to our hospital because of a right paranasal painless swelling, which appeared 9 months ago and was slowly increasing in size since her first visit. A Magnetic Resonance Imaging (MRI) revealed a tumor (18 × 25 × 37 mm) centered on the right ethmoidal cells, with adjacent nasal bone erosion and a right orbital invasion. The tumor was removed with a combined surgical approach (endoscopic and trans-sinusal frontal approach) after pre-surgical embolization. Based on the results of the histopathological examination, the diagnosis of a glomangiopericytoma was confirmed. The patient was discharged after one week, and an uneventful long-term follow-up.

## 1. Introduction

Glomangiopericytomas are a rare type of sinonasal tumors [1]. They were first described in 1942 by Stout and Murray [2]. The etiology is not well understood. These tumors are sporadic and there are no known genetic or environmental risk factors for their development [3]. They are characterized as a pile of capillary blood vessels with the presence of pericytes. These cells enable the synthesis of collagen fibers around capillaries, thereby allowing contraction by transforming them into smooth muscles. These tumors mostly affect the sphenoid and ethmoid sinuses [1].

The endoscopic approach is the classical technique for the removal of this type of sinonasal lesions [4]. The orbital invasion is infrequently described in the literature and Jung et al. first described a case of glomangiopericytoma extension in the orbital cavity [5]. Here we report a case of sinonasal glomangiopericytoma with orbital extension that required a combined surgical approach to remove the tumor and reconstruct the bony defect.

## 2. Presentation of Case

The authors confirm that informed consent was obtained from the patient for this publication.

A 50 year-old woman was referred to our hospital because of a right paranasal painless swelling, which appeared 9 months ago and was slowly increasing in size since her first visit. The patient described a nasal obstruction predominantly on the right side. The patient did not have any visual disturbance. The 2 cm right paranasal swelling was indurated but not fixed to the skin. An anterior rhinoscopy showed a right septal deviation, as well as the tip of a smooth mass medially to the middle turbinate. An MRI showed a tumor centered on the right ethmoidal cells, with adjacent bone erosion (nasal bone) and the start of a right orbital invasion. The CT-scan revealed a blown aspect of the lateral wall of the ethmoidal sinus with an extension to the paranasal fat (Figure 1). The imaging suggested a highly vascularized benign tumor. We therefore opted for surgery because of the large size, localization of the tumor and, also the orbital invasion. To minimize the risk of bleeding, a pre-surgical embolization of the lesion was performed the day of the surgery. The patient was put under general anesthesia for the whole duration of the surgery. Embolization is performed by direct puncture and injection of liquid embolization material (4 cc of precipitating hydrophobic injectable liquid [PHIL^®^] Tustin, CA, USA) and resorbing Gelfoam^®^ (New-York, NY, USA) into both maxillary arteries (Figure 2). The surgical approach was first a coronal approach with a trans-sinusal frontal approach. Then, the anterior part of the right frontal sinus was removed with burr to access the tumor (Figure 3). A major part of the tumorectomy was performed by this approach and was completed by an endocopic endonasal procedure. This second approach allowed removal of tumor residue located at the level of the ethmoidal sinus and the skull base. The anterior part of the right frontal sinus was osteosynthesized with mini plates and 4 mm mini screws (Synthes^®^, Johnson & Johnson, Warsaw, IN, USA) (Figure 4). The bone defect (medial orbital wall, and nasal bone) was reconstructed with a titanium mesh fixed with 4 mm mini screws (Synthes^®^). Two drains were placed in the detachment space (subperiosteal in the frontal region and susperiosteal in the parietal region), sutures were performed layer by layer (Vicryl 3.0, Ethicon^®^, Johnson & Johnson, Warsaw, IN, USA) for deeper planes and Prolene 3.0 (Ethicon^®^). A nasal packing was performed for 5 days. Classical postoperative care was prescribed: Amoxicillin and clavulanic acid for 7 days, analgesics, skin and endonasal care. The histological examination revealed a sinonasal glomangiopericytoma with a low mitotic index (Figure 5). No further treatment was required and the patient was discharged from the hospital eight days after the surgery. No complication occurred and the patient did not suffer from diplopia. Postoperative CT-scan confirmed the complete removal of the tumor and showed the titanium plate reconstructing the orbito-nasal bone defect (Figure 6). The patient benefited from a radio-clinical follow-up with a control MRI at 1, 3 and 6 months (post-operatively) showing no tumor recurrence. The radio-clinical follow-up by MRI was thereafter repeated every 6 months.

## 3. Discussion

Glomangiopericytomas are a type of sinonasal tumors with high recurrence rates, rarely described in the literature. The principal symptoms are pain (72% of cases) and swellings (69% of cases), often with a nasal obstruction similar to our patient [6]. Average size is 3 cm [7]. These tumors have a very slight female predominance. All ages can be affected, but the peak is in the 7th decade [8]. Differential diagnostics of these lesions can vary widely and must include hemorrhagic nasal mass as glomus tumors, angiofibromas, lobular capillary hemangiomas, angioleiomyoma [1]. Radiological images are not sufficient to establish strong evidence for a diagnosis, which thus remains histopathological.

Histopathology shows a richly vascularized proliferation consisting of uniform spindle cells with hyperchromatic nuclei and poorly delineated eosinophilic cytoplasm, forming clusters and storiform structures like in our case. The mitotic index is usually low and no necrosis is found. Inflammatory cells, myxoid change and red blood cells can also be observed [9,10].

The local recurrence rate is high after surgical treatment and reach up to 40%. Multiple recurrences can happen in 5–10% of cases. In case of high recurrence rates, this low malignant potential tumor can metastasize [1]. These recurrences are induced mainly due to insufficient resection of the mass and there is no consensus recommendation as per its clinical management in the literature [8]. Preoperative treatment by embolization may also be considered to facilitate excision and prevent bleeding [1,11,12,13,14]. However, one author doesn’t use embolization but cauterization during surgery and intranasal packing, with an estimated blood loss of 20 cc [11]. One author does not specify the surgical technique nor the blood loss [15]. All articles agree on a surgical treatment, often via an endoscopic approach. In order to avoid recurrence, the choice of a surgical approach is crucial [12,16]. In our case, the tumor was invading both the right nasal bone and the medial wall of the orbit. As radiotherapy is not possible near the eye, a combined approach was therefore performed to not only remove the tumor but also reconstruct the bone lysed by the tumor. Trans-sinusal frontal approach was particularly appropriate to visually control the complete excision of the tumor during surgery and perform bone reconstruction [17]. The choice of titanium mesh was made with two functions in mind: first, to restore the volume of the orbit, and second, to ensure the impermeability between the nasal cavity and the orbit. It is worth noting that titanium meshes have the disadvantage to induce more severe facial traumas in case of shocks in comparison to resorbable plates, which are also less expensive [18]. In this case, the three-dimensional conformation of the bone defect did not allow reconstruction with resorbable material. This type of approaches allows the restoration of facial symmetry and ocular function without visible scare in the face (only on the scalp but hidden by the hair). In the case of metastases, chemotherapy and radiotherapy should be considered.

## 4. Conclusions

Sinonasal glomangiopericytomas are extremely rare tumors. They require surgical management, and the approach must be adapted to the location and the extension of the tumor. Preoperative imaging is necessary to plan the approach according to the location and extension of the mass. The choice of the surgical approach is crucial to control the resection of the mass, thus avoiding recurrences. Long-term follow-up is necessary for such high-potential recurrence tumors.

## Figures and Tables

**Figure 1 reports-06-00007-f001:**
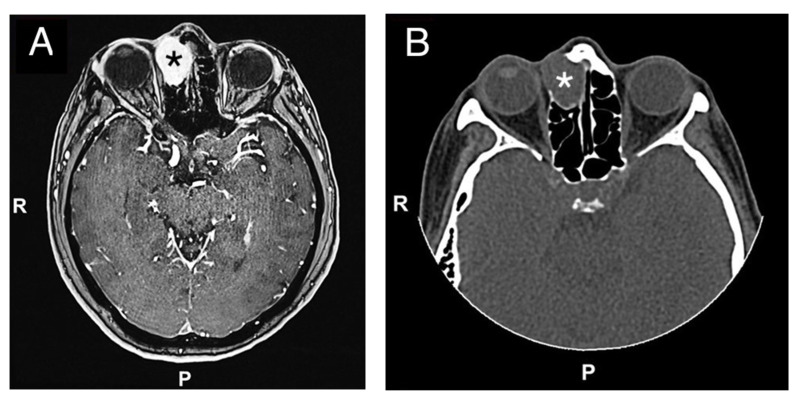
(**A**) Preoperative MRI (Time Of Flight TOF sequence in axial view; black asterisk) and CT-scan (**B**) (axial view; white asterisk) (**C**) (coronal view; white asterisk), (**D**) (sagittal view; white asterisk) showing an anterior ethmoidal high vascularized tumor invading the right nasal bone and the medial wall of the right orbit. R = right, P = posterior.

**Figure 2 reports-06-00007-f002:**
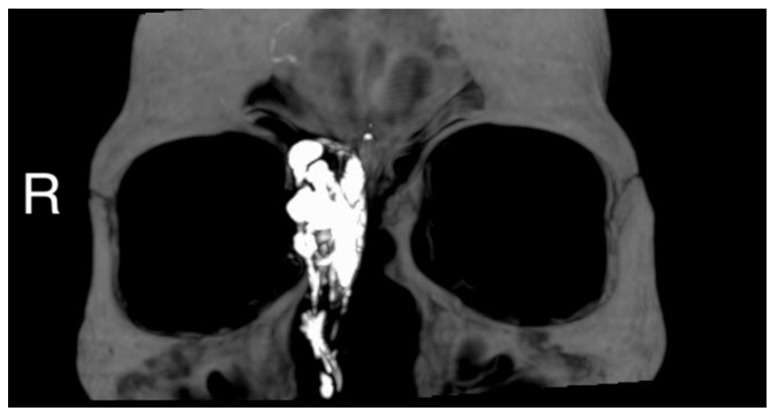
Visualization of the embolized mass in white (3D view). R = right.

**Figure 3 reports-06-00007-f003:**
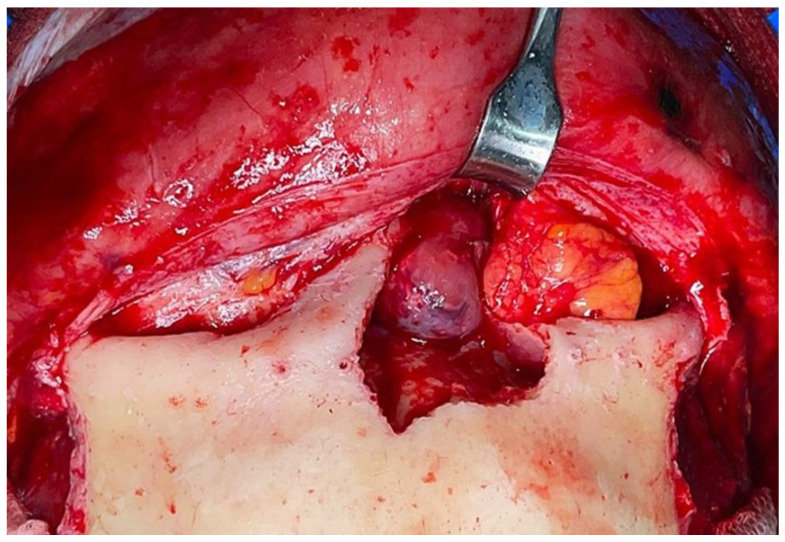
Intraoperative view of the tumor through a trans-sinusal frontal approach.

**Figure 4 reports-06-00007-f004:**
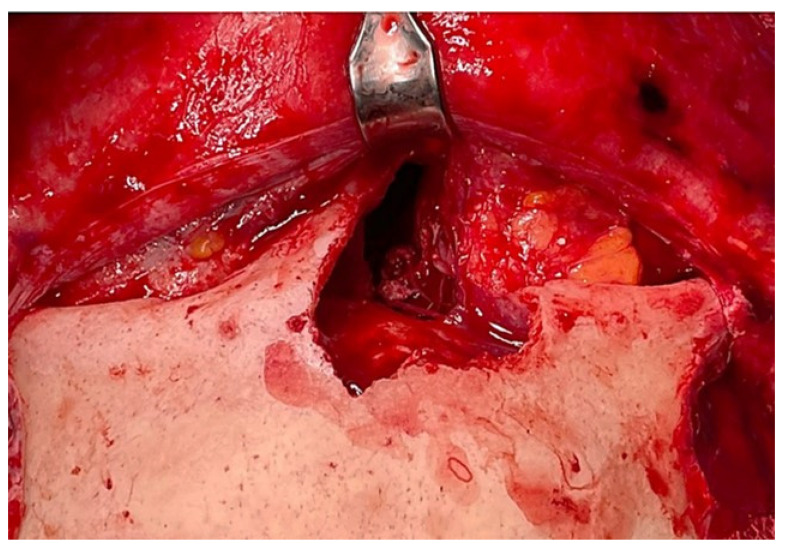
Intraoperative view after complete removal of the tumor.

**Figure 5 reports-06-00007-f005:**
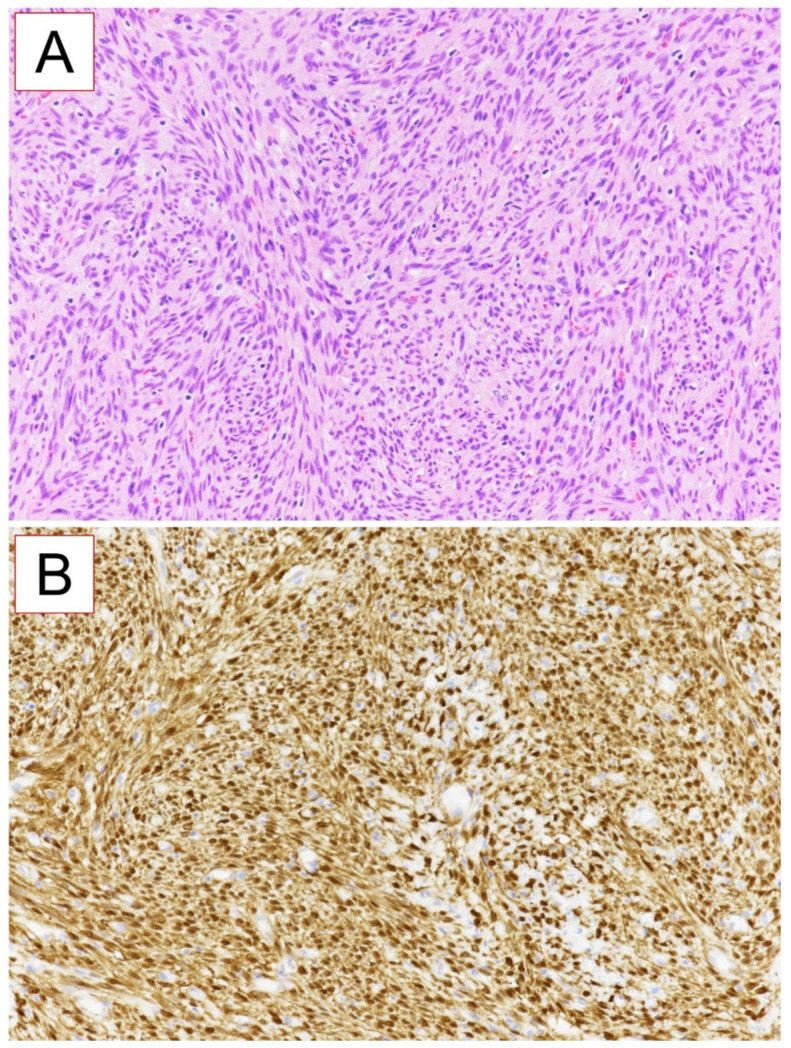
Histological examination of glomangiopericytoma: HE coloration (**A**) and β-catenin immunostaining (**B**) (×200 magnification) shows richly vascularized proliferation consisting of uniform spindle cells with hyperchromatic nuclei and poorly delineated eosinophilic cytoplasm, forming clusters and storiform structures.

**Figure 6 reports-06-00007-f006:**
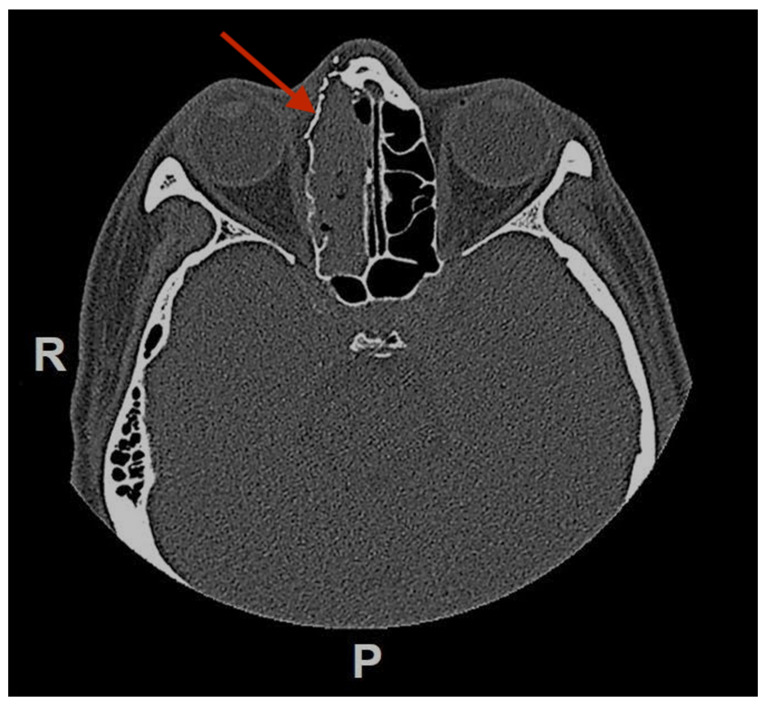
Postoperative (3 weeks) CT-scan in axial view showing the reconstruction of the medial right orbit wall with a titanium mesh. (red arrow pointing at the titanium mesh). R = right, P = posterior.

## Data Availability

The data presented in this study are available on request from the corresponding author. The data are not publicly available due to privacy.

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
