# Peer review of "A Combined Surgical Approach for the Treatment of a Sino-Naso-Orbital Glomangiopericytoma"

_reports, 2023, doi:10.3390/reports6010007_

Round 1

Reviewer 1 Report

Dear authors,

This article is generally well written and it can contribute to the literature. But it needs some adjustments,

Suggested arrangements;

- English grammar language editing required

- In the discussion part, more emphasis should be placed on the advantageous and disadvantageous parts of this surgery.

Bes regards...

Author Response

Dear Reviewer,

Thank you for your feedback and comments.

R1: English grammar language editing required

ALA: Thank you for your feedback and comments.

The English language has been revised.

R: In the discussion part, more emphasis should be placed on the advantageous and disadvantageous parts of this surgery.

ALA: We opted for surgery because of the large size, localization of the tumor and, also the orbital invasion (L93-95). In our case, the tumor was invading both the right nasal bone and the medial wall of the orbit. As radiotherapy is not possible near the eye, a combined approach was therefore performed to not only remove the tumor but also reconstruct the bone lysed by the tumor. (L.207-210)

We modified the text accordingly

We hope that these corrections will suit you.

Best regards

Reviewer 2 Report

Thank you for giving me the chance to evaluate the article. This paper is focused on an interesting topic, and it may be interesting for the readers. I want to compliment with the Authors for the work done and for the excellent iconography.Efficiently described case, in the future I would suggest adding a literature review. Then the work will be more extensive and will contribute more to the scientific approach to the subject.

Author Response

Dear Reviewer,

Thank you for your positive feedback.

As this article was intended as a case report, we opted for a more direct approach with a focused list of references. We have expanded our literature based on your comments.

We hope that these corrections will suit you.

Best regards,

Reviewer 3 Report

This is a case report but, as such, You must discuss the embolization : when (in view of operation time) did You do this embolization, and how (in with vessel and with what kind of drug )? The discussion must also be done about the interest (vs risk..) of this embolization...

The use of a Ti-plate to reconstruct the orbital wall must also be discussed : first, this plate is not visible on the postoperative scan view, and second, why didn't you use a resorbable plate ? This should at least be discussed

L. 37 : paitent should be patient

L. 39 : was there a first visit of the patient to your clinic ? and if so, when and what were the complaints of the patient at this moment ?

The references should be harmonized in their presentation..

Author Response

R3: This is a case report but, as such, You must discuss the embolization : when (in view of operation time) did You do this embolization, and how (in with vessel and with what kind of drug )? The discussion must also be done about the interest (vs risk..) of this embolization...

ALA: Thank you for your feedback and comments. We have adapted the article according to your comments:

The patient was put under general anesthesia. Successive serigraphies were performed in the right internal and external carotid arteries. A hypervascular blush of tumor-like appearance was found in the right nasal fossa fed almost exclusively by ethmoidal afferents from the right ophthalmic artery arising from the right internal carotid artery. Embolization of the lesion was performed by directly puncturing it with a 22-gauge needle via the transnasal route, under fluoroscopic control, followed by the injection a liquid embolization material (precipitating hydrophobic injectable liquid) PHIL® 30 and then PHIL® 25 for a total volume of about 4 cc. On control, a total devascularization of the lesion with reflux of embolization material into the afferent arteries was observed, respecting the orbital vascularization on the right and in particular the retinal artery with a good choroidal blush. In addition, a resorbing Gelfoam® was injected in the two maxillary arteries on both sides (Fig. 2). The control in the internal carotid arteries on both sides showed a preserved patency of the intracerebral arterial network. We finally removed the material and closed the puncture site by manual compression and a safeguard inflated with 50cc of air (deflation of 50% after 2 hours and removal after 4 hours). (L96-110)

Preoperative treatment by embolization may also be considered to facilitate excision and prevent bleeding (1,11–14). However, one author doesn’t use embolization but cauterization during surgery and intranasal packing, with an estimated blood loss of 20cc (11). One author does not specify the surgical technique nor the blood loss (15). (L202-205)

We have also added a 3D image of the embolization (L 315)

R3: The use of a Ti-plate to reconstruct the orbital wall must also be discussed: first, this plate is not visible on the postoperative scan view, and second, why didn't you use a resorbable plate? This should at least be discussed

ALA: The CT scan image was modified with a red arrow to indicate the location of the mesh. (L 328)

In agreement with your comment, we have added the justification of a titanium mesh in the Discussion section. The choice of titanium mesh was made with two functions in mind: first, to restore the volume of the orbit, and second, to ensure the impermeability between the nasal cavity and the orbit. It is worth noting that titanium meshes have the disadvantage to induce more severe facial traumas in case of shocks in comparison to resorbable plates, which are also less expensive (18). In this case, the three-dimensional conformation of the bone defect did not allow reconstruction with resorbable material. This type of approach allows the restoration of facial symmetry and ocular function without visible scarring in the face (only on the scalp but hidden by the hair). (L.211-218)

R3: L. 37 : paitent should be patient

ALA: the typo was corrected

R3: L. 39 : was there a first visit of the patient to your clinic ? and if so, when and what were the complaints of the patient at this moment ?

ALA: at the beginning of section 2 "Presentation of case", we refer to the first visit of the patient and her symptoms: A 50-year-old woman was referred from the department of Ophthalmology to our department of Otorhinolaryngology, Head and Neck Surgery and Maxillofacial Surgery. The patient complained of a right paranasal swelling, which appeared 9 months ago and was slowly increasing in size since her first visit in the department of Ophthalmology. This evolutive paranasal tumefaction was painless. The patient described a nasal obstruction predominantly on the right side and a periorbital discomfort. The patient did not present diplopia and did not have any visual disturbance. (L63-68):

R3: The references should be harmonized in their presentation..

ALA: we have harmonized their presentation.

We hope the above will answer your questions and concerns.

Reviewer 4 Report

The manuscript entitled “A Combined Surgical Approach for the Treatment of a Sino-Naso-Orbital Glomangiopericytoma” is a case report about a glomangiopericytoma that involved the orbital cavity. The tumor itself is already described in the literature. However, the involvement of the orbital cavity is indeed rare, with only fell reports.

The case is interesting and well-illustrated. However, some information are missing and the theme itself could be further developed, especially in a journal that there is no maximum number of words nor limits of images/references number.

Introduction: 

Quite short. Goes from basic cellular characterization directly to the treatment. I suggest the authors developing a bit more of the context of the disease before describing the treatment, that is, etiology, prevalence and frequent areas of involvement (sphenoid and ethmoid sinuses). In addition, the authors could add alternative treatments besides surgery, if an attempt was performed (even though they did not work).

Presentation of the case:

 Pre-operative: Detailed, with symptoms, CT imaging and diagnostics. Ok

Trans/post-operative: Describe the incisions, general and local anesthesia

Add the brand/system of titanium plates/mesh/screws, as well as the suture material (s).  

Add the location of the insertion of drains.

Describe the technique employed to perform the osteotomy (burrs, Piezo, osteotome).

Would be illustrative the inclusion of an image of the Ti plate inserted prior sutures.

Being the orbital cavity involved, did the patient develop diplopia? The information should be described, regardless the answer.

I suggest adding a symbol or an arrow pointing out the tumor in CT images (it is obvious for someone familiar to TC images; however, I believe that might help people from other backgrounds to understand the images).  

The CT images inserted in the manuscript are only axial. Other planes would be illustrative.

Figure 4: 200x magnification is quite high for optical microscopy. Would be interesting to see the tissue at lower magnifications as well.

Being a high recurrence lesion, describe the follow-up plan for the patient – frequency and method to identify possible recurrences.

Discussion:

The authors cite the Differential diagnostics of glomangiopericytoma. I suggest some clarification regarding the differences among these lesions, from histology to clinical, explaining how to differentiate them, etc.

Despite being rare, it is possible to find case reports about glomangiopericytoma affecting the orbital cavity. Would be very useful and enriching to compare some characteristics of the present study with the others in the literature.

Ethics:

Compliance with Ethical Standards (Helsinki Declaration), consent to participate and consent for publication must be disclosed by the authors. The first phrase of the presentation of case addresses an informed consent. However, patient must consent about the publication as well, even though her identity is not revealed by the pictures.

Author Response

The manuscript entitled “A Combined Surgical Approach for the Treatment of a Sino-Naso-Orbital Glomangiopericytoma” is a case report about a glomangiopericytoma that involved the orbital cavity. The tumor itself is already described in the literature. However, the involvement of the orbital cavity is indeed rare, with only fell reports.

The case is interesting and well-illustrated. However, some information are missing and the theme itself could be further developed, especially in a journal that there is no maximum number of words nor limits of images/references number.

ALA: Thank you for your feedback and comments. We have adapted the article according to your comments:

Introduction: 

R4: Quite short. Goes from basic cellular characterization directly to the treatment. I suggest the authors developing a bit more of the context of the disease before describing the treatment, that is, etiology, prevalence and frequent areas of involvement (sphenoid and ethmoid sinuses). In addition, the authors could add alternative treatments besides surgery, if an attempt was performed (even though they did not work).

ALA: Glomangiopericytomas (sinonasal-type hemangiopericytomas) are a very rare type of sinonasal tumors arising from the pericytes surrounding capillaries (accounts for less than 0.5% of all sinonasal tumors) (1). They were first described in 1942 study by Stout and Murray based on nine different cases (2). The etiology is unfortunately not well understood. These tumors are sporadic and there are no known genetic or environmental risk factors for their development (3). They are characterized as a pile of capillary blood vessels with the presence of pericytes. These cells enable the synthesis of collagen fibers around capillaries, thereby allowing contraction by transforming them into smooth muscles. These tumors mostly affect the sphenoid and ethmoid sinuses (1). (L47-54)

Presentation of the case:

R4:  Pre-operative: Detailed, with symptoms, CT imaging and diagnostics. Ok

R4: Trans/post-operative: Describe the incisions, general and local anesthesia

ALA: The patient was put under general anesthesia. Successive serigraphies were performed in the right internal and external carotid arteries. A hypervascular blush of tumor-like appearance was found in the right nasal fossa fed almost exclusively by ethmoidal afferents from the right ophthalmic artery arising from the right internal carotid artery. Embolization of the lesion was performed by directly puncturing it with a 22-gauge needle via the transnasal route, under fluoroscopic control, followed by the injection a liquid embolization material (precipitating hydrophobic injectable liquid) PHIL® 30 and then PHIL® 25 for a total volume of about 4 cc. On control, a total devascularization of the lesion with reflux of embolization material into the afferent arteries was observed, respecting the orbital vascularization on the right and in particular the retinal artery with a good choroidal blush. In addition, a resorbing Gelfoam® was injected in the two maxillary arteries on both sides (Fig. 2). The control in the internal carotid arteries on both sides showed a preserved patency of the intracerebral arterial network. We finally removed the material and closed the puncture site by manual compression and a safeguard inflated with 50cc of air (deflation of 50% after 2 hours and removal after 4 hours).

The patient was put under general anesthesia for the whole duration of the surgery and antibiotic prophylaxis was administered (amoxicillin/clavulanic acid 2.2g intravenous.). After shampooing and braiding of the hair to delimit the bicoronal approach, the skin was disinfected with an antiseptic (povidone-iodine cream). A classical septoplasty infiltration was performed with a mixture of Lidocaine-Carbostesin-Adrenalin 1%. The procedure was performed under neuronavigation. The surgical approach was first a coronal approach with a trans-sinusal frontal approach. A scalp incision was made from one ear to the other (bicoronal) with a cold scalpel. The incision was continued to the supra-periosteal plane, still using the scalpel (section of the galea). Hemostasis was performed with bipolar forceps for the larger subcutaneous vessels and Raney clips were placed along both edges of the incision on the scalp. The dissection between the galea and the periosteum was continued to the frontal region. In front of the suture between the frontal bone and the two temporal bones, the periosteum was incised with a scalpel and then a subperiosteal detachment with a periosteal elevator. After exposure of the supraorbital region, the nasal region was released of the right and left supraorbital nerves. The bone window to be performed in the anterior table of the frontal sinus was delineated using neuronavigation. The anterior part of the right frontal sinus was removed with burr to access the tumor (Fig. 3). The bone was conserved in saline solution. Fluid retention in the frontal sinuses was observed. A major part of the tumor was shown through this surgical approach and an incomplete tumorectomy was performed. During the tumorectomy, we took care not to enter the sieve blade. The tumorectomy was completed by an endoscopic endonasal procedure. We first opened the ethmoidal bulla and then all the ethmoidal cells up to the anterior wall of the sphenoid sinus. We then continued to open all the ethmoidal cells up to the base of the skull and the frontal sinus. A small effraction of the periorbital fat was noted at the site of lysis of the right papyraceous blade. This second approach allowed removal of tumor residue located at the level of the ethmoidal sinus and the skull base. The procedure was completed with a septoplasty due to the fact that the patient had a significant anterior septal deviation, which would greatly limit postoperative rhino care. A right hemi-transfixing incision was made, boring upper and lower tunnels. A watch glass quadrangular cartilage was noted on the right side, inferior and posterior chondrotomy. A resection of an inferior part of the quadrangular cartilage was performed, leaving the height anteriorly for the projection. We also resected a part of the vomer and the perpendicular blade of the ethmoid. The height of the premaxilla was reduced. After these different maneuvers, the septum was finally straightened. After the visually complete removal of the tumor, there was a large communication between the right orbit and the nasal cavity. (L96-144)

(…)

Classical postoperative care was prescribed: Amoxicillin and clavulanic acid for 7 days, analgesics, skin and endonasal care standard rhino-care. The removal of the wicks was made 24 hours after surgery, while the pressure dressing was removed 48 hours after the surgery. Ten days after the operation, Nasopore was sucked out and silastics were removed. (L153-157)

R4: Add the brand/system of titanium plates/mesh/screws, as well as the suture material (s).  

ALA: The anterior part of the right frontal sinus was reinstated and was osteosynthesized with two mini plates and 4mm mini screws (Synthes®, Johnson & Johnson, Warsaw, IN, USA) (Fig. 4). The bone defect (medial orbital wall, and nasal bone) was reconstructed with a conformed on-site titanium mesh fixed with 4mm mini screws (Synthes®). Two drains were placed in the detachment space (subperiosteal in the frontal region and susperiosteal in the parietal region), sutures were performed layer by layer (Vicryl 3.0, Ethicon®, Johnson & Johnson, Warsaw, IN, USA) for deeper planes and Prolene 3.0 (Ethicon®). Closure was done with Monosyn Quick 5.0. We placed trans-columellar stitches of Monosyn Quick 4.0., two silastics (1mm) fixed with Prolen 4.0, Nasopore in place of the lumpectomy and finally two Tampograss wicks. A pressure dressing was applied with Velpo bandage. (L144-153)

R4: Add the location of the insertion of drains.

ALA: Two drains were placed in the detachment space (subperiosteal in the frontal region and susperiosteal in the parietal region) (L148-149)

R4: Describe the technique employed to perform the osteotomy (burrs, Piezo, osteotome).

ALA: the periosteum was incised with a scalpel and then a subperiosteal detachment with a periosteal elevator. After exposure of the supraorbital region, the nasal region was released of the right and left supraorbital nerves. The bone window to be performed in the anterior table of the frontal sinus was delineated using neuronavigation. The anterior part of the right frontal sinus was removed with burr to access the tumor (Fig. 3). (L123-127)

R4: Would be illustrative the inclusion of an image of the Ti plate inserted prior sutures.

ALA: Sadly, we don’t have images of the titanium mesh. The approach and soft tissues did not allow satisfactory iconography of the mesh in place before closure

R4: Being the orbital cavity involved, did the patient develop diplopia? The information should be described, regardless the answer.

ALA: No complication occurred and the patient did not suffer from diplopia. Immediate postoperative CT-scan confirmed the complete removal of the tumor. Also, it showed the titanium mesh reconstructing the orbito-nasal bone defect and the correct position of the material (Fig. 6). (L170-173)

R: I suggest adding a symbol or an arrow pointing out the tumor in CT images (it is obvious for someone familiar to TC images; however, I believe that might help people from other backgrounds to understand the images).  

ALA: Asterisks were added where the tumor is located. (L306)

R4: The CT images inserted in the manuscript are only axial. Other planes would be illustrative.

ALA: Coronal and Sagittal planes were added. (L307)

R4: Figure 4: 200x magnification is quite high for optical microscopy. Would be interesting to see the tissue at lower magnifications as well.

ALA: Sadly, we don’t have any other histological images.

R4: Being a high recurrence lesion, describe the follow-up plan for the patient – frequency and method to identify possible recurrences.

ALA: The patient benefited from a radio-clinical follow-up with a control MRI at 1, 3 and 6 months (post-operatively) showing no tumor recurrence. The radio-clinical follow-up by MRI was thereafter repeated every 6 months. (173-175)

Discussion:

R4: The authors cite the Differential diagnostics of glomangiopericytoma. I suggest some clarification regarding the differences among these lesions, from histology to clinical, explaining how to differentiate them, etc.

ALA: We opted for a surgical approach with a focus on the dual procedure (coronal and endoscopic). We nonetheless duly noted your suggestion, and we will implement this in future works.

R4: Despite being rare, it is possible to find case reports about glomangiopericytoma affecting the orbital cavity. Would be very useful and enriching to compare some characteristics of the present study with the others in the literature.

ALA: To our knowledge, there are only few reports of such glomangiopericytoma affecting the orbital cavity. We have nonetheless included them to our discussion and as additional references.

Ethics:

R4: Compliance with Ethical Standards (Helsinki Declaration), consent to participate and consent for publication must be disclosed by the authors. The first phrase of the presentation of case addresses an informed consent. However, patient must consent about the publication as well, even though her identity is not revealed by the pictures.

ALA: We have added an ethical section “Ethics” (L.241-242) confirming that the patient has consented to the publication of her case.

Round 2

Reviewer 3 Report

Thanks to the authors for corrections and modifications of their manuscript

Reviewer 4 Report

The authors significantly improved the manuscript, following the suggestions of reviewers and replying our questions.

Therefore, this case report is now suitable for publication. Congratulations.